# Carlo.jl: A general framework for Monte Carlo simulations in Julia

**Lukas Weber**[1,2,⋆]

**1** Center for Computational Quantum Physics, The Flatiron Institute, 162 Fifth Avenue, New York, New York 10010, USA

**2** Max Planck Institute for the Structure and Dynamics of Matter, Luruper Chaussee 149, 22761 Hamburg, Germany

⋆ [lweber@flatironinstitute.com](mailto:lweber@flatironinstitute.com)

## Abstract

**Carlo.jl is a Monte Carlo simulation framework written in Julia. It provides MPI-parallel scheduling, organized storage of input, checkpoint, and output files, as well as statistical postprocessing. With a minimalist design, it aims to aid the development of high-quality Monte Carlo codes, especially for demanding applications in condensed matter and statistical physics. This hands-on user guide shows how to implement a simple code with Carlo.jl and provides benchmarks to show its efficacy.**

# 1 Introduction

Monte Carlo methods have been a powerful computational tool since the very beginnings of the fields of statistical and condensed matter physics [1]. Initially built to study the classical Boltzmann distribution, they perform well on a wide variety of high-dimensional integration problems due to their ability to break the curse of dimensionality. In quantum systems, where a probabilistic interpretation of the partition function is in general not possible, quantum Monte Carlo algorithms can exploit certain quantum-to-classical mappings to make powerful and accurate predictions for models that are out of reach of other state-of-the art methods [2].

In exchange for simulating high-dimensional problems, quantum Monte Carlo simulations in particular rely on efficient implementations making use of massive parallelism. In the course of a simulation, many random samples are produced that have to be stored and statistically postprocessed to arrive at results with reliable errorbars. These requirements typically necessitate some amount of "bookkeeping" code, unrelated to the Monte Carlo algorithm itself, but still crucial to its performance. This inspired the development of earlier frameworks such as the ones included in ALPS [3] and the subsequent ALPSCore [4] in C++.

The Julia programming language [5] is becoming increasingly popular as an alternative to traditional scientific computing languages such as C++ and Fortran. Its heavy reliance on type inference, dynamic dispatch, and just-in-time compilation allow it to provide both high performance and the flexibility of a high-level scripting language. Furthermore, its modern approach to dependency management greatly facilitates installing and reusing existing software.

The purpose of Carlo.jl is to provide a high-performance Monte Carlo framework for Julia. Its features include (i) Monte Carlo aware parallel scheduling (ii) organized and reproducible storage of simulation inputs, checkpoints, and results, as well as (iii) statistical postprocessing of autocorrelated Monte Carlo samples.

In Section 2, the features of Carlo.jl and their inner workings are explained in detail. Section 3 showcases how to implement a Monte Carlo code in Carlo.jl using the paradigmatic Metropolis Monte Carlo algorithm for the Ising model as an example. Finally, in Section 5, a short conclusion and outlook are given.

Apart from this user guide, users may refer to the Carlo.jl documentation[1] for more detailed descriptions of each component.

# 2 Features

The purpose of this section is to give a high-level overview of the way Carlo.jl works without descending too far into technical details and concrete usage instructions. For a more hands-on introduction into how to use Carlo.jl in practice, see Section 3.

## 2.1 Monte Carlo aware parallel scheduling

One useful property of Monte Carlo simulations is that they are embarrassingly parallel. An arbitrary number of random processes can be allocated to many workers in parallel, which, once thermalized, can all produce samples that are finally averaged together. Apart from this intrinsic parallelism, simulations are often run for a variety of different parameters in parallel. Both of these kinds of parallelism can be exploited using a standard parallel scheduler. However, if the load between different parameter sets is not balanced, the resource usage of this approach is not optimal: Most workers can run out of work while few are still stuck

---

[1]The documentation can be viewed online on GitHub, built locally by running `cd docs; julia --project make.jl`, or accessed directly within the Julia REPL.

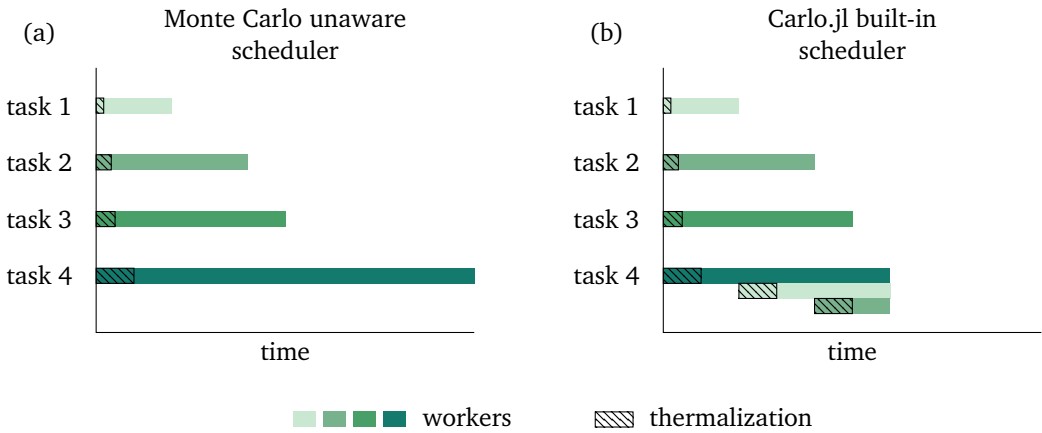

Figure 1: **Parallelism in Monte Carlo simulations**. (a) Monte Carlo unaware parallelization over different parameters (e.g. system sizes, temperatures) can lead to suboptimal resource usage if tasks are not well balanced. (b) Carlo.jl parallelizes over different parameter sets but also takes advantage of the possibility to share work between different Monte Carlo processes for the same task. Each new process needs to thermalize first, which is a serial cost that can only be reduced by more fine-grained parallelization.

simulating the last few expensive parameter sets (Fig. 1(a)). In Monte Carlo simulations, such a situation can always be avoided by simply allowing the idle workers to join other tasks and share the work (Fig. 1(b)).

Carlo.jl uses MPI.jl [6] to orchestrate this kind of Monte Carlo aware, work-sharing scheduling and automatically averages the results. Each parameter set in the simulation corresponds to a *task* and each task may have multiple *runs*, which correspond to independent random processes that are averaged. If workers are idle, the scheduler will assign them a new task from the set of unfinished tasks in a round-robin manner and start a new run for that task, provided the remaining work is not less that the number of thermalization sweeps.

On top of this model of trivial parallelism, Carlo.jl also supports nontrivial MPI parallelism. The work of each run may optionally be shared by more than one MPI rank. This advanced feature, called *parallel run mode*, can be helpful in situations where the runtime of a simulation is dominated by the thermalization steps. It also allows the natural implementation of branching random walk algorithms on top of Carlo.jl. While the Ising example will not use this feature, further information on how to use parallel run mode is available in the documentation.

## 2.2 Organized storage of inputs, checkpoints and results

As mentioned in the previous section, large-scale Monte Carlo simulations can involve scanning large parameter spaces and produce large numbers of random samples. Long-running simulations should also implement some form of checkpointing so that valuable computation time is not wasted when an error occurs or the job runs out of its allocated runtime on a computing cluster.

These boundary conditions induce data management challenges at the beginning, during, and at the end of a simulation. This section explains how Carlo.jl handles these challenges for the user.

### 2.2.1 Input

Rather than specifying simulation parameters in traditional configuration files, Carlo.jl leverages the dynamic nature of Julia to set parameters programmatically. The recommended workflow for this is to write a job script as shown in Listing 1. The job script can be split into three parts. First, the `TaskMaker` from `Carlo.JobTools` is used to procedurally generate a list of parameter sets. Then, this list is retrieved using `make_tasks(tm)` and fed into the `JobInfo` alongside other options such as the maximum runtime limit and the checkpoint interval. Finally, `start(job, ARGS)` initiates the Carlo.jl command line interface (CLI) when the job script is executed.

```julia
# example_job.jl

using Carlo
using Carlo.JobTools
using Ising

tm = TaskMaker()

tm.sweeps = 20000
tm.thermalization = 2000
tm.binsize = 100

Ts = range(1, 3, 10)
Ls = [8, 12, 16]
for L in Ls
    for T in Ts
        tm.T = T
        tm.Lx = L
        tm.Ly = L
        task(tm)
    end
end

job = JobInfo("example_job", Ising.MC;
    run_time = "24:00:00",
    checkpoint_time = "30:00",
    tasks = make_tasks(tm),
)
start(job, ARGS)
```

Listing 1: **Carlo job script**. This snippet generates a job for the Ising code implemented in Section 3. Arbitrary parameters can be assigned to the `TaskMaker` `tm`. The function `task(tm)` creates a snapshot of all currently set values and turns it into a task that will be simulated. The list of tasks is returned by `make_tasks` and passed to the `JobInfo` structure. The detailed meanings of the options are explained in Section 3.

The CLI allows *running* the simulation, and for unfinished simulations allows checking their *status* and *merging* the already available data to obtain preliminary results.

While recommended, the use of `TaskMaker` and the CLI is optional. For more information on how to manually start simulations from within Julia, interested readers are referred to the documentations of `JobInfo` and the `start` function.

### 2.2.2 Checkpointing

In a high-performance computing environment, calculations often run on limited time allocations that may be shorter than the length of the calculation one wishes to perform. If a calculation runs out of time or uses too much memory, it may be terminated by the environment, which leads to the loss the entire progress.

Checkpointing, i.e. saving the complete Monte Carlo configuration to disk at regular intervals, allows to gracefully recover from such situations and simply continue the calculation from the point it left off. Carlo provides a simple interface to save the configuration state to an HDF5 [7, 8] group. Internally, it makes sure that these checkpoints are never corrupted, even if the process is terminated in the middle of writing.[2] Running a simulation as described in the previous subsection will automatically resume from checkpoints if they exist.

While packages like JLD2.jl [9] exist that allow automatically saving arbitrary Julia objects, the Carlo.jl checkpointing interface instead relies on manually choosing which data to save in what way. This can be helpful in reducing the checkpoint file size and encourages maintaining stable serialization interfaces.

### 2.2.3 Results

Like checkpoints, Carlo.jl stores simulation results in HDF5 files. Inspecting these files allows retrieving the entire Monte Carlo time series. In the postprocessing step that is described in Section 2.3, these raw files are merged and averaged to produce means, statistically correct error bars and approximate autocorrelation times. These merged results are saved in a human-readable JSON file. The `Carlo.ResultTools` module provides a simple way to import these JSON files into a Julia Dataframe (e.g. from `DataFrames.jl` [10]).

To facilitate reproducibility, and conform with the best practices of research data management, the result files produced by Carlo.jl contain all parameters used to run the simulation as well as the version of Carlo.jl and of the package containing the Monte Carlo algorithm implementation.

## 2.3 Postprocessing

Monte Carlo time-series are in general autocorrelated. Carlo.jl performs postprocessing on measured observables to provide the user with statistically sound means and standard errors using a binning analysis.

The binning analysis works in two steps. First, an *internal binning* step is performed during the simulation before Monte Carlo measurements are written to disk. The bin size for this step is set by `binsize` in the task parameters. The purpose of internal binning is simply to reduce memory and disk-I/O overhead. As long as `binsize` is not big compared to the total number of samples, it does not influence the magnitude of the final errors computed by Carlo.jl.

Once the simulation is complete, for a given task, there will be multiple files containing measurement results from different independent runs that were scheduled by Carlo.jl (Fig. 1(b)). In the second step of the binning analysis, these internally binned Monte Carlo time series are concatenated and binned further. The purpose of this "rebinning" step is to produce sufficiently large bins to recover uncorrelated statistics that allow extracting the statistical errors using the standard estimator

$$\sigma_A = \sqrt{\frac{1}{N_{\text{rebin}}(N_{\text{rebin}} - 1)} \sum_n (\bar{A}_n - \bar{A})^2}, \tag{1}$$

---

[2]This works by writing to a temporary file first and then moving that file to the final location, overwriting the old file. Moving (as opposed to writing) is an atomic operation that cannot be interrupted.

where $\bar{A}_n$ are the means of each rebinned sample and $\bar{A} = \sum_n \bar{A}_n / N_{\text{rebin}}$. By default, the number of "rebins" $N_{\text{rebin}}$ is chosen using the heuristic

$$N_{\text{rebin}}(N) = N_{\text{rebin}}^{\min} + \max\{0, \sqrt[3]{N - N_{\text{rebin}}^{\min}}\} \tag{2}$$

where $N$ is the total number of measured bins and $N_{\text{rebin}}^{\min} = 10$ is a minimum in case there are only few bins in the simulation. Choosing a cube-root scaling is a simple way to ensure that both $N_{\text{rebin}}$ and the length of each rebin, $N/N_{\text{rebin}}$, approach infinity for increasing $N$, while devoting more resources to the latter. The default value of $N_{\text{rebin}}$ can be overriden using the task parameter `rebin_length` in the job script.

A rough estimate for the autocorrelation time $\tau_{\text{auto}}$ is calculated based on the ratio of the rebinned error $\sigma_A$ and naive error computed without any rebinning, $\tilde{\sigma}_A$, [11]

$$\tau_{\text{auto}} = \frac{\sigma_A^2}{2\tilde{\sigma}_A^2}. \tag{3}$$

Note that $\tau_{\text{auto}}$ is measured in units of the internal bin size (set by the `binsize` parameter). To find the autocorrelation time in terms of Monte Carlo sweeps, `binsize` should be set to 1.

Apart from the observables themselves, users can also define *evaluables* that depend on the observables via some function. The bias-corrected means and errors for evaluables are then propagated using the jackknife [12] method on the rebinned samples.

## 3 Usage example: Ising model

In this section, we will give a hands-on usage example for Carlo.jl by implementing and running the Metropolis Monte Carlo algorithm [1] for the Ising model on a square lattice,

$$H = -\sum_{x,y} \sigma_{x,y}^z (\sigma_{x+1,y}^z + \sigma_{x,y+1}^z), \tag{4}$$

with periodic boundary conditions. All code for this example is available in a separate repository [13].

### 3.1 Implementation

First, we need to implement the algorithm. It is a good idea to write such an implementation as its own Julia package. In the following, we will assume that we have generated a package called `Ising`, with a file `Ising.jl` containing all the code.

In this file, we declare a new subtype of `AbstractMC`.

```julia
# Ising.jl
module Ising

using Carlo
using HDF5

struct MC <: AbstractMC
    T::Float64

    spins::Matrix{Int8}
end
```

The type MC holds two fields: the temperature T and a matrix of integers representing the spins on the square lattice.

The constructor of MC should accept a dictionary of parameters that are defined in the jobscript (see Listing 1).

```
function MC(params::AbstractDict)
    Lx = params[:Lx]
    Ly = get(params, :Ly, Lx)
    T = params[:T]
    return MC(T, zeros(Lx, Ly))
end
```

We read the temperature and lattice dimensions from the parameters and construct the MC structure accordingly.

The remainder of the implementation consists of methods for Carlo.jl's AbstractMC interface.

```
function Carlo.init!(mc::MC, ctx::MCContext, params::AbstractDict)
    mc.spins .= rand(ctx.rng, Bool, size(mc.spins)) .* 2 .- 1
    return nothing
end
```

The Carlo.init! function is called to initialize the Monte Carlo configuration at the beginning of a new simulation (but not when restarting from a checkpoint).

In addition to the MC object, it receives a MCContext, which is our handle to interact with Carlo.jl's functionality. Here, we use the integrated random number generator ctx.rng to generate a spin configuration. It is important to always use ctx.rng for random numbers to maintain reproducibility from a given random seed.[3]

Next we implement the Monte Carlo updates or sweeps.

```
function periodic_elem(spins::AbstractArray, x::Integer, y::Integer)
    return spins[mod1.((x, y), size(spins))...]
end

function Carlo.sweep!(mc::MC, ctx::MCContext)
    Lx = size(mc.spins, 1)

    for _ = 1:length(mc.spins)
        i = rand(ctx.rng, eachindex(mc.spins))
        x, y = fldmod1(i, size(mc.spins,1))

        neighbor(dx, dy) = periodic_elem(mc.spins, x + dx, y + dy)
        ratio = exp(
            -2.0 / mc.T *
            mc.spins[x, y] *
            (neighbor(1, 0) + neighbor(-1, 0)
            + neighbor(0, 1) + neighbor(0, -1)),
        )

        if ratio >= 1 || ratio > rand(ctx.rng)
            mc.spins[x, y] *= -1
        end
    end
    return nothing
end
```

---

[3]The seed can be set in the jobscript via the parameter seed.

According to the Metropolis algorithm, we propose a spin flip and calculate the energy difference between the new and old configurations, which can be reduced to a calculation of the neighboring spins. Note the use of functions like mod1 and fldmod1, which are helpful when applying modular arithmetic to the 1-based indices of Julia arrays.

Next are the measurements.

```julia
function Carlo.measure!(mc::MC, ctx::MCContext)
    mag = sum(mc.spins) / length(mc.spins)

    energy = 0.0
    correlation = zeros(size(mc.spins, 1))

    for y = 1:size(mc.spins, 2)
        for x = 1:size(mc.spins, 1)
            neighbor(dx, dy) = periodic_elem(mc.spins, x + dx, y + dy)
            energy += -mc.spins[x, y]*(neighbor(1,0)+neighbor(0,1))

            # in practice, one should use more lattice symmetries!
            correlation[x] += mc.spins[1, y] * mc.spins[x, y]
        end
    end

    measure!(ctx, :Energy, energy / length(mc.spins))

    measure!(ctx, :Magnetization, mag)
    measure!(ctx, :AbsMagnetization, abs(mag))
    measure!(ctx, :Magnetization2, mag^2)
    measure!(ctx, :Magnetization4, mag^4)

    measure!(ctx, :SpinCorrelation, correlation ./ size(mc.spins, 2))
    return nothing
end
```

We calculate the magnetization and its moments, the energy per spin, and the spin correlation function. The function measure!(ctx, name, value) passes each result to Carlo.jl. As we see for the correlation function, observables need not be scalars. Vectors (or higher-rank) arrays are also supported.

The next method is optional, but can be used to register *evaluables*, that is, quantities that depend on Monte Carlo expectation values. Carlo.jl will perform error-propagation and bias correction on these (see Section 2.3).

```julia
function Carlo.register_evaluables(
        ::Type{MC}, eval::Evaluator, params::AbstractDict
)
    T = params[:T]
    Lx = params[:Lx]
    Ly = get(params, :Ly, Lx)

    evaluate!(eval, :BinderRatio,
        (:Magnetization2, :Magnetization4)
    ) do mag2, mag4
        return mag2 * mag2 / mag4
    end

    evaluate!(eval, :Susceptibility, (:Magnetization2,)) do mag2
        return Lx * Ly * mag2 / T
    end
```

```
evaluate!(eval, :SpinCorrelationK, (:SpinCorrelation,)) do corr
    corrk = zero(corr)
    for i = 1:length(corr), j = 1:length(corr)
        corrk[i] += corr[j] * cos(2pi/length(corr)*(i-1)*(j-1))
    end
    return corrk
end

return nothing
end
```

Using `evaluate!`, we provide formulas for useful quantities such as the Binder ratio, the susceptibility and the Fourier transform of the correlation function. Note that `Carlo.register_evaluables` is called during the postprocessing step, after the simulation is complete. It has no access to any simulation state.

Finally, we implement the checkpointing interface.

```
function Carlo.write_checkpoint(mc::MC, out::HDF5.Group)
    out["spins"] = mc.spins
    return nothing
end

function Carlo.read_checkpoint!(mc::MC, in::HDF5.Group)
    mc.spins .= read(in, "spins")
    return nothing
end

end # module Ising
```

When starting a simulation from a checkpoint, `Carlo.read_checkpoint!` is called instead of `Carlo.init!`, so that we have to recover everything from disk that is not already calculated in the constructor `MC(params)`. The temperature is set in the constructor so we do not have to save it to disk. The spin configuration, however, has to be saved. For more information on how to read and write to an `HDF5.Group`, see the documentation of HDF5.jl [7].

This concludes the `Ising.jl` file and our implementation. Next, we will use it to run some simulations.

## 3.2  Running simulations

In an environment, where we have access to the `Ising` module,[4] we write the job script from Listing 1. The meanings of the parameters are as follows.

`Lx, Ly, T` used by our implementation to set the size and temperature of the Ising model.

`thermalization` the number of sweeps that are performed in the thermalization phase of the simulation, before measurements are taken.

`sweeps` the number of measurement sweeps.

`binsize` the internal bin size, that is the number of samples that are averaged before writing to disk. This should be much smaller than the total number of sweeps but large enough to not waste too much disk space in large simulations.

---

[4]For example, we can add the module to environment using `]dev path/to/Ising` in the Julia REPL.

There are more optional parameters that are explained in greater detail in the documentation for `TaskInfo` and `JobInfo`.

Using the given system sizes and temperatures, we generate a collection of tasks. Then, all information about the job is fed to the `JobInfo` constructor, which additionally takes the job name, here `"example_job"`, the Monte Carlo type we implemented, `Ising.MC`, as well as the total runtime and the interval between checkpoints.

Finally, `start(job, ARGS)` invokes the Carlo.jl CLI, which we can run from the shell as follows.

```
$ julia example_job.jl --help
$ julia example_job.jl run
```

The first line gives an overview over the CLI, the second line starts the simulation. For long-running simulations, the commands `status` and `merge` are helpful to check the progress and postprocess those results that are already available, respectively.

Once the simulation completes, next to the job script, we should see a directory `example_job.data`, which contains raw data and check points, and a file `example_job.results.json`, which contains the postprocessed results of our simulation. The submodule `Carlo.ResultTools` provides tools for reading these results back into Julia.

```julia
using Plots
using DataFrames
using Carlo.ResultTools

df = DataFrame(ResultTools.dataframe("example_job.results.json"))
plot(df.T, df.BinderRatio;
    xlabel = "Temperature",
    ylabel = "Binder ratio",
    group = df.Lx,
    legendtitle = "L",
)
```

The output of this snippet can be seen in Fig. 2.

To inspect a raw Monte Carlo time series, we can read it directly the respective HDF5 file. For example, for the first temperature and system size, we could do

```julia
using HDF5

samples = h5read(
    "example_job.data/task0001/run0001.meas.h5",
    "observables/Energy/samples",
)
```

to retrieve a vector containing each recorded sample of the energy.

In the example above, we ran Carlo.jl with a single processor. In order to use the parallel scheduler, we can simply launch the Carlo.jl CLI with MPI.

```
$ mpirun -n $ncores julia example_job.jl run
```

If the simulation has already been completed before, this will stop without doing further work. To restart the simulation, we should use the flag `run -r` or call the `delete` command before running.

In production runs, it is advisable to configure MPI.jl and HDF5.jl to use the system-provided binaries.

```julia
using MPIPreferences
MPIPreferences.use_system_binary()
```

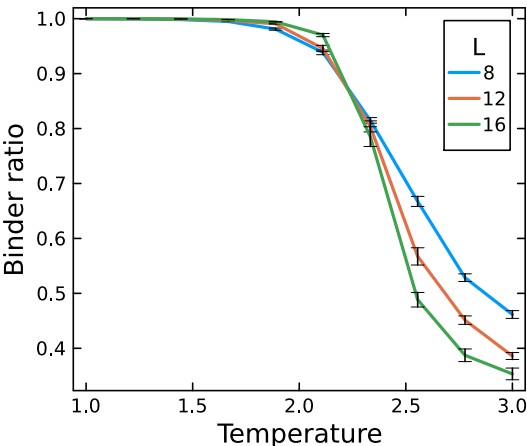

Figure 2: **Binder ratio**, $\langle m^2 \rangle^2 / \langle m^4 \rangle$, of the magnetization $m$, in the $L \times L$ square-lattice Ising model as a function of temperature. As one would expect, there is a crossing close to the critical temperature $T_c \approx 2.269$. The error bars are calculated automatically by Carlo.jl using the jackknife method.

```
using HDF5
HDF5.API.set_libraries!(
    "path/to/libhdf5.so",
    "path/to/libhdf5_hl.so",
)
```

Further, by default, Julia will use multithreading for linear algebra operations. Without care, together with MPI parallelization, this can cause overusage of the available CPUs, so it should be controlled, e.g. by setting the environment variable `OPENBLAS_NUM_THREADS=1`.

## 4  Benchmarks

In this section, we will provide benchmarks to show (i) the runtime scaling of the parallel scheduler and (ii) the validity of the statistical postprocessing in the face of strongly autocorrelated data. For both, we will use the Ising example code implemented in Section 3.

For the runtime scaling, we consider a workload inspired by Listing 1, where we make the replacements `Ls = [100, 200, 300, 500]` and `sweeps = 200000` to increase the computational effort. Since the Ising code scales quadratically with the linear system size `Lx`, the tasks of this workload are quite imbalanced, comparable to the situation in Fig. 1. Nevertheless, we observe nearly linear scaling of the runtime with number CPUs used (Fig. 3). We expect that the deviation from perfect scaling in this case is dominated by the serial cost of thermalization, which, based on the number of sweeps, is about 1% of the total serial runtime.

To confirm the accuracy of the statistical postprocessing, we consider the Ising model with $L = 20$ and $T = 2.3$, close to the critical point. In this region, the local updates of the Metropolis algorithm that we implemented are quite inefficient, leading to long autocorrelation times that require a careful analysis to retrieve the correct statistics.

To obtain the statistics of this model, there are two ways. The first one is to run many identical, independent simulations and create a histogram of the results. The second, more practical approach is to run a single simulation and perform a binning analysis on the correlated time series. By comparing the histogram of many independent simulations with the

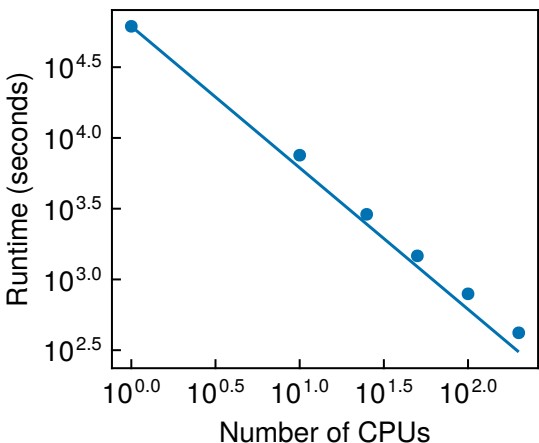

Figure 3: **Runtime scaling with number of CPUs** of the example job script from Listing 1, with the replacements Ls = [100, 200, 300, 500] and sweeps = 200000. The solid line shows the ideal linear speedup.

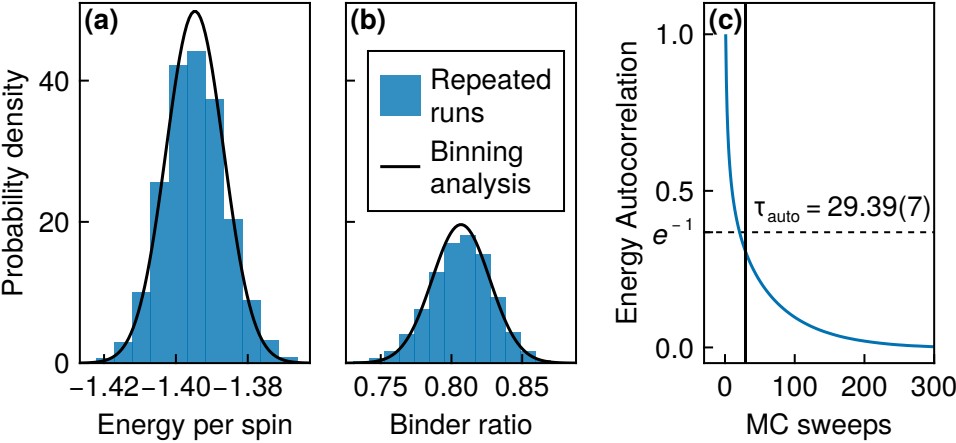

Figure 4: **Binning analysis benchmark** using the $20 \times 20$ square-lattice Ising model at temperature $T = 2.3$, close to the critical point. The same simulation with 20000 sweeps and 4000 thermalization sweeps is repeated 9000 times. We then compare the distribution of these results to the distribution predicted by the binning analysis. (a) Histogram of the energy per spin compared to a Gaussian distribution with the average mean and standard deviation predicted from the binning analysis. (b) Histogram of the Binder ratio compared to a Gaussian distribution with the average mean and standard deviation predicted from jackknifing. (c) The energy autocorrelation function, averaged over all runs, compared to the autocorrelation time $\tau_{\mathrm{auto}}$ predicted by Carlo.jl.

average results from the binning analysis implemented in Carlo.jl (Fig. 4(a)), we confirm that the binning procedure outlined in Section 2.3 yields the correct errors for observables. As an example for the jackknifing feature, the Binder ratio (Fig. 4(b)) also shows correct statistics. Finally, in Fig. 4(c) we show the autocorrelation function for the energy per spin. We find that the autocorrelation time estimated from the binning analysis using Eq. (3) is a good estimate for the true decay of the autocorrelations.

## 5    Conclusion

Carlo.jl is a framework for writing Monte Carlo simulations in Julia. It features a Monte Carlo aware parallel scheduler, organized storage of input, checkpoint, and result files, as well as statistical postprocessing of the Monte Carlo results.

We have illustrated these features and the general usage of the framework by providing an example implementation of the Metropolis algorithm for the Ising model. Further benchmarks, based on the example implementation have been provided to show the runtime scaling of the scheduler and the correctness of the statistical postprocessing.

Based on these features and properties, the availability of a framework like Carlo.jl should enable the quick development of new user-friendly high-performance Monte Carlo codes in Julia.

## Acknowledgements

Carlo.jl is inspired by an earlier C++ code [14] written by the author, which in turn was inspired by a legacy code used in Stefan Wessel's group. All plots except Fig. 2 were created using Makie.jl [15].

**Funding information**    L.W. acknowledges support by the Deutsche Forschungsgemeinschaft (DFG, German Research Foundation) through grant WE 7176-1-1. The Flatiron Institute is a division of the Simons Foundation.

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
