# Peer review of "Carlo.jl: A general framework for Monte Carlo simulations in Julia"

_SciPost Physics Codebases_

## Round 1 · Referee Report · Anonymous (Referee 1) · 2024-9-30

Strengths

1- Provides a practical scheduler to optimize CPU usage 2- Includes a basic set of typical measurements that can be used

Weaknesses

1- Only one simple example is provided 2- Requires a large effort for the user, such that it may only be used by experts

Report

The article introduced Carlo.jl, a Julia package that helps implement Monte Carlo methods to different problems by providing the basic functions generally needed and a task scheduler that allows to comprise all desired runs into a single script. First, it is nice to see that Julia packages are being programmed, documented, and published. The theoretical physics community is stearing in this direction and, since Julia is a rather new language, there is much room to develop and share open code.

As a Monte Carlo user, there is a point that was unclear when first reading the article. Is Carlo.jl aimed for quantum or classical Monte Carlo calculations? It later became clear that Carlo.jl only provides the core functions needed for a battery of applications, but the extent of applications should be made immediately clear to the reader. This is just a minor and personal issue.

About the code, I see the advantage of using Carlo.jl. However, since most of the model programming depends on the user, the package seems to be appealing to experts only. To say it bluntly, it is probably too much work for someone who needs a quick result to compare with either experiments or some other theory. This could be greatly alleviated with the consistent inclusion of more examples of application (not in the article, but in the library). To say something: Heisenberg, XY, Ising, classical, quantum, chain, square, triangular, cubic. Something to ease the way into Monte Carlo methods for beginners.

Otherwise, people who have already been working on Monte Carlo, probably have their code. And I find it hard to believe that someone would exchange their own code, where one knows exactly how it works and how to adapt it to each scenario, for some other code where they still need to code in some complicated model.

I recognize that this is a too personal point of view, and I cannot predict how popular the package will be. My recommendations are merely suggestions to widen the scope of posible users. That being said, I see no reason why the code (and article) should not be published as is. Ideally, we (scientists) should stop having our own codes for everything, wasting time in programming the same things over and over. Especially when they can be generalized. And this package goes in that direction.

Recommendation

Publish (meets expectations and criteria for this Journal)

  • validity: good
  • significance: good
  • originality: good
  • clarity: high
  • formatting: excellent
  • grammar: excellent

Author:  Lukas Weber  on 2024-11-26  [id 4995]

(in reply to Report 1 on 2024-09-30)

I thank the reviewer for their positive evaluation of our manuscript and the recommendation to publish it without modifications.

It is true that the scope of Carlo.jl is limited to implementers of Monte Carlo algorithms, rather than a broader audience. The referee is correct in that this scope should have been stated more clearly from the beginning. Therefore, several formulations in the introduction have been changed for clarification.

I agree that adding more open source examples, especially of state-of-the-art research codes, would make Monte Carlo methods more accessible to new users. However, I believe Carlo.jl should concentrate on solving one specific problem well: being a reusable Monte Carlo framework. Instead of being a part of it, application codes using Carlo.jl should exist separately from it. As a first step, for this revision, I have published my state-of-the-art stochastic series expansion quantum Monte Carlo code and included it as an example in the user guide for using Carlo.jl nontrivially.

---

## Round 1 · Referee Report · Anonymous (Referee 2) · 2024-10-10

Strengths

  • Simple and well-written Julia package for performing Monte Carlo simulations

Weaknesses

  • Lack of applications (e.g. different stat-mech models, quantum Monte Carlo) that would make users change practice. Ising application is too simple. To have impact, this scheduler/error analysis library needs to come with a more complete software environment
  • Little added value of the paper accompanying the code

Report

This paper is associated to a Julia library designed to help perform Monte Carlo simulations. The library contains a scheduler (to perform computations in parallel, including with checkpointing) as well as error analysis. Those are the basic blocks of every Monte Carlo simulation.

I am not a Julia practitioner, and actually couldn’t install the package (but that’s probably due to my lack of knowledge of the Julia environment). I am nevertheless convinced that the package works and is efficient.

It somehow fills a gap that was left by the C++ Alps library (somewhat popular in this community, but no longer maintained) which contained similar features but also many other features (lattice. Model libraries) and application packages, and another previous attempt in Julia (MonteCarlo.jl for which the status in unclear).

Now an important question remains: what is the target audience for this package / paper package ? I suspect this is for Monte Carlo practitioners (mostly in the field of statistical mechanics and quantum (lattice) models and I tentatively see two possible specific target groups : 1. Monte Carlo practitioners who have not switched to Julia (: these researchers have probably their own Monte Carlo frameworks already, and it is not obvious that this paper will make them switch to Julia for the following reasons : the Ising model example is too simple, the performances (e.g. good naive parralelization in Figure 3) are certainly identical to those with their own house-made Monte Carlo, and there are no new specific non-trivial Monte Carlo aspects that are introduced in this package (see suggestion below however) 2. Julia practitcionners that may need to do Monte Carlo simulations. I see perhaps an interest for this category.

I naturally think that the readership of SciPost Physics Codebases is mostly composed of researchers in category 1 (this is my case). Let me also mention that, as a matter of fact, the paper accompanying the code is very simple, and not much more instructive than a well-made tutorial that could be included in the library.

Based on this, I think that there is perhaps more to be done to convince (me and the rest of the readership in this category 1) that this library is an added value that merits a new publication be included in the SciPost Physics Codebases. At the moment, I don’t think the simple port to Julia of standard Monte Carlo features is enough. As suggestions for improvements, I recommend to :

  • Document and explain using a non-trivial example of what is the parallel run mode, which promises to offer nontrivial MPI parallelism
  • Provide a more advanced applicative package (than just Ising, that is unlikely to be very useful), e.g. in quantum lattice models, that is based on this Monte Carlo Julia library. The author has publications using quantum Monte Carlo simulations with Stochastic Series Expansions (SSE), and the documentation on the github mentions quantum Monte Carlo SSE and auxiliary field codes, based on the framework. I think this would be an added value for which category 1 audience could cross the Julia Rubicon.

Requested changes

  1. Provide at least one more advanced applicative package (than just Ising, that is unlikely to be very useful), e.g. in quantum lattice models, that is baed on this Monte Carlo Julia library. see main report for suggestions

  2. Document and explain using a non-trivial example of what is the parallel run mode

Recommendation

Ask for major revision

  • validity: good
  • significance: ok
  • originality: ok
  • clarity: ok
  • formatting: reasonable
  • grammar: good

Author:  Lukas Weber  on 2024-11-26  [id 4994]

(in reply to Report 2 on 2024-10-10)

I am thankful for the critical evaluation of our manuscript. The referee finds that Carlo.jl “somehow fills a gap that was left by the C++ Alps library” and a previous attempt in Julia. They identify two possible target audiences: (1) Monte Carlo practitioners who have not switched to Julia and (2) Julia practitioners that may need to do Monte Carlo.

While they see interest for category 2, the referee comes to the conclusion that category 1 needs additional arguments to use Carlo.jl. As examples they suggest to explain the parallel run mode and to add a more advanced applicative package.

To explain the parallel run mode feature, I have provided an implementation of parallel tempering. Since parallel tempering requires the synchronization of multiple Monte Carlo simulations in parallel, it usually requires nontrivial changes to the core of the parallel scheduling logic. However, in Carlo.jl, as shown in the revised manuscript, parallel tempering is straight-forward to implement on top of parallel run mode.

Not only does this flexibility in implementing nontrivial parallelism set Carlo.jl apart from the average in-house code, but parallel tempering itself is a useful, often missed feature that can now benefit Carlo.jl simulations.

For the second point of adding a more advanced application package, I have included the example of StochasticSeriesExpansion.jl, which is a state-of-the-art Julia implementation of the stochastic series expansion quantum Monte Carlo method. While explaining the inner workings of this code is beyond the scope of the user guide, a practical calculation example that can be followed by non-specialists to reproduce some experimental results is included.

Further, since StochasticSeriesExpansion.jl is a direct port of an earlier C++ code, which was used in the publications mentioned by the referee, it presents an opportunity for a careful performance comparison between Julia and C++ for quantum Monte Carlo applications. This comparison is included in the resubmission for the broader interest of the community. In the comparison, the Julia version runs slightly faster than the C++ version. Since both codes can likely be optimized further, this does not prove that one language is intrinsically faster. However, it shows that Julia can be used to write Monte Carlo codes that reach the performance of state-of-the-art C++ codes.

Together, these two additional examples may present a small bridge over the Julia rubicon for some researchers of category 1. Since the referee agrees that the features of Carlo.jl are part of every Monte Carlo simulation, I believe that this package can significantly lower the cost of testing and adopting Julia. According to the engagement on Github, it has already found such early adopters, just based on the arXiv submission.

In addition to the changes mentioned above, more detailed installation instructions have been added.

---

## Round 2 · Referee Report · Anonymous (Referee 2) · 2024-12-12

Strengths

  • Simple and well-written Julia package for performing Monte Carlo simulations

  • A simple (Ising code) and more complex (stochastic series expansion, a quantum monte calro method) example are now provided

Report

The main two points in my previous reported (explanation of parallel mode, addition of a new appliative package) have been clearly taken into account. by the author.
The corresponding new features (parallel tempering and stochastic series expansion code) are clear added values to the code and thus broadne the range of potential users of the package.

Requested changes

There are very minor points to fix : - The abstract has not been changed to reflect the new additions - Minor : I am not sure the sign problem in the model Eq 5 comes only from the signs of J_ij . Couldn't there be sign problem due to sign of D_i^x ? - A final reading could be useful to improve the style, in particular of the newly added paragraphs (e.g. 'Now, we will now sketch ...')

Recommendation

Publish (easily meets expectations and criteria for this Journal; among top 50%)

  • validity: high
  • significance: good
  • originality: good
  • clarity: good
  • formatting: good
  • grammar: reasonable

Author:  Lukas Weber  on 2024-12-13  [id 5039]

(in reply to Report 1 on 2024-12-12)

We thank the referee for their careful reevaluation of the manuscript.

The statement about the sign-problem was indeed imprecise. The sign-problem free case is when the sign of $D^x_i$ is uniform throughout the lattice. The reason is that the off-diagonal component of $(S^x)^2$, consisting $(S^+)^2$ and $(S^-)^2$ can only appear in even numbers due to the magnetization conservation of the remaining Hamiltonian.
Therefore, there will always be an even power of $D^x_i$ operators and an even number of extra $J_{ij} S^+_i S^-_j$ paths connecting them, leaving the sign positive on a bipartite lattice.
The above argument breaks however, when $D^x_i$ has different signs on different lattice sites. A discussion of this has been added.

Apologies for the typos. After adjusting the abstract to incorporate the changes, I have checked the manuscript to improve the style.

---

## Editorial Decision

resubmitted